# Prompt Recovery for Image Generation Models: A Comparative Study of Discrete Optimizers

## Abstract

Recovering natural language prompts for image generation models, solely based on the generated images is a difficult discrete optimization problem. In this work, we present the first head-to-head comparison of recent discrete optimization techniques for the problem of *prompt inversion*. Following prior work on prompt inversion, we use CLIP's Radford et al. (2021) text-image alignment as an inexpensive proxy for the distribution of prompt-image pairs, and compare several discrete optimizers against BLIP2's image captioner (Li et al., 2023) and PRISM (He et al., 2024) in order to evaluate the quality of discretely optimized prompts across various metrics related to the quality of inverted prompts and the images that they generate. We find that while the discrete optimizers effectively minimize their objectives, CLIP similarity between the inverted prompts and the ground truth image acts as a poor proxy for the distribution of prompt-image pairs – responses from well-trained captioners often lead to generated images that more closely resemble those produced by the original prompts. This finding highlights the need for further investigation into inexpensive methods of modeling the relationship between the prompts for generative models and their output space.

## 1 Introduction

Images generated by AI models are flooding the internet and the models and prompts used for generation often feel like components of alchemy. Naturally, the academic community aims to better understand the mechanisms at play in these systems, with one crucial focus on inverting the generative process by recovering prompts from images (Fan et al., 2024).

As the name suggests, given only the output image from a generative model, prompt inversion methods seek to find the prompt that generated the image. While the goal is clear, this problem leads to a wide variety of approaches. For example, given an initial prompt estimate, by strategically refining the text input Sohn et al. (2023) have found that the image generation process can be effectively controlled. Yet, this refining process can be difficult, leading to Wen et al. (2023) leveraging CLIP's (Radford et al., 2021) embedding space to directly optimize natural language inputs to be close to target images. Similarly, Mahajan et al. (2024) have proposed an inversion technique that backpropagates through intermediate steps of the diffusion process instead of relying on CLIP embeddings In contrast, it has been found that training a captioner on a dataset of prompt-image pairs effectively learns the prompt distribution well enough to act as an inverter.

While prompt inversion is an interesting task in its own right, there are two practical motivations to develop strong methods. First, those interested in better controlling the output of image generation models may want to find a prompt from an image as a starting point for their own prompt engineering. Second, by extracting prompts from images, one might better understand the various associations these text-to-image models have and debug them to avoid explicit content. Yet, to date, there is no standardized comparison of these methods for prompt inversion from image generation models.

In this work, we benchmark several approaches to prompt inversion and attempt to shed light on three primary questions: 1) How closely do the distribution of images generated by each approach align with the distribution generated by the original prompt? 2) Do text-image alignment models like CLIP successfully act as a proxy for the joint distribution of prompt-image pairs? 3) Does the ability for discrete optimization methods to search the input space allow them to outperform the ingrained knowledge of learned models? We investigate these questions by comparing four generic methods of discrete optimization within the prompt space (Zou et al., 2023; Wen et al., 2023; Zhu et al., 2023; Andriushchenko, 2023) against two methods that directly sample from the space (Li et al., 2023; He et al., 2024) and evaluate each across several metrics. We follow our analysis with a discussion on the efficacy of different approaches to approximating and searching within prompt spaces.

## 2 Related Work

To best situate this paper among prior work, we discuss several motivations for executing prompt inversion, other domains where discrete optimization is relevant, and the overall goal of our work in contrast to related papers.

Here, we focus on discrete optimization methods for recovering image prompts, but discrete optimization over natural language has several other applications including jailbreaking LLMs (Andriushchenko, 2023; Zou et al., 2023; Zhu et al., 2023) and measuring memorization (Schwarzschild et al., 2024; Kassem et al., 2024). In particular, Zou et al. (2023) propose a method to find adversarial prompts for LLMs that break their safety alignment. We experiment with this optimizer for image generation inversion as Zou et al. (2023) compare their optimizer to PEZ in their work, but only with the goal of jailbreaking LLMs.

Whereas prompt optimization strategies in the text generation space have specific goals, such as generating targeted strings, the image generation space has struggled with tractable options for aligning prompts and generated images. While we follow prior work in using CLIP as a proxy model, this choice is primarily driven by the practical challenges of directly optimizing prompts through backpropagation in the diffusion process. The computational requirements for coarse-grained exploration and fine-grained search that discrete optimization often calls for (Parker and Rardin, 2014) would entail generating multiple full images for every candidate prompt att every step. Following the search parameters recommended by (Zou et al., 2023), optimizing a single prompt would require generating a minimum of 512 images per step, which quickly becomes prohibitive over numerous iterations.

Mahajan et al. (2024) have attempted to address this burden by focusing on the similarity between predicted noise residuals at specific diffusion timesteps, rather than generating full images. However, in alignment with prior work on noise inversion (Song et al., 2020; Mokady et al., 2023) the authors find that prompts only have strong influence on the generated image during a narrow range of timesteps. At early timesteps, the image becomes largely "locked in," so even substantial changes to the prompt have little effect. In contrast, at later timesteps, the stochasticity of the diffusion process leads to large variations in the final image, even when the correct prompt is used. This unpredictability makes it difficult to rely on noise residual comparisons for consistent prompt inversion. Thus, prompt inversion methods (Wen et al., 2023; Williams and Kolter, 2024) often rely on deterministic proxy models like CLIP, which offer a more stable and efficient alternative. Through CLIP's text-image alignment, we can more reliably approximate the prompt-image distribution without having to address the risk of stochasticity producing divergent results.

Importantly, direct discrete optimization is not the only method for finding viable prompts. Several approaches focus on using black box models to sample prompts. Both Zhang et al. (2024) and He et al. (2024) use pretrained language models to extract prompts for given a output across text generation and image generation tasks respectively. Moreover, as we show in this work, even a simple captioner that has not been finetuned for prompt generation often outperforms discrete optimization methods. In fact, Reade et al. (2023) have found that a captioner fine-tuned on pairs of prompts and the images that they generate can effectively sample prompts that are exceptionally similar to the ground truth.

Despite this performance, we focus on the discrete case as direct discrete optimization can be beneficial for better understanding the behavior of the image generation models. Similarly to prior work on counterfactual explanations (Verma et al., 2020), directly optimizing inputs for a desired outputs helps to better understand the decision boundaries of classifiers (Ribeiro et al., 2016). While generative models are not classifiers with explicit decisions and accuracy metrics, they are constantly making decisions on their representations based on the prompts. From background color to subject ethnicity, discrete optimization methods may provide a useful understanding of the relationship between prompts and images (Williams et al., 2024).

We emphasize solidifying ways of comparing discrete optimizers for image generation tasks. Even with the rise of novel discrete optimization methods, standard prompt recovery comparisons over images are missing. We focus on a holistic benchmark on not only the similarity between prompt and image, but also the similarity among images generated by the inverted prompts which to the best of our knowledge has not been standardized in this setting.

## 3   Selected Algorithms

To best introduce the optimizers we study, it is critical to pose the prompt inversion problem formally. Consider a tokenizer that maps from natural language to sequences of integer valued tokens corresponding to a list of indices in a vocabulary of tokens $\mathbb{T}$. Let $x \in \mathbb{T}^s$ be a length $s$ sequence of tokens. Next, let $\mathbf{E} \in \mathbb{R}^{|\mathbb{T}| \times d}$ be a matrix whose rows are $d$-dimensional embedding vectors, one for each token in the vocabulary. To embed a sequence $x$, we can define $\mathbf{X} \in \{0,1\}^{s \times |\mathbb{T}|}$ s.t. $\sum_{i=1}^{T} \mathbf{X}_{j,i} = 1 \ \forall j \in \{1, ..., M\}$ to be a matrix of one-hot encoded rows for the integers in the sequence $x$. The product $\mathbf{XE}$ defines an $s \times d$ embedding of $x$.

Prompt inversion techniques seek to find the sequence of tokens $x$, or equivalently their corresponding one-hot encodings $\mathbf{X}$, that solve $\mathcal{M}^{-1}(Y)$, where $\mathcal{M}$ is a stochastic generative model that maps a sequence of tokens $x$ to an image $Y$. Typically we express the solution as the minimizer of some loss function $\mathcal{L}$, or the solution to the following optimization problem.

$$\operatorname*{argmin}_{\mathbf{X} \in \{0,1\}^{s \times |\mathbb{T}|}} \quad \mathcal{L}(\mathcal{M}(\mathbf{XE}), Y) \quad \text{s.t.} \quad \sum_{i=1}^{|\mathbb{T}|} X_{j,i} = \mathbf{1} \ \forall \ j \in \{1, ..., s\} \tag{1}$$

Khashabi et al. (2021) show that embeddings in $\mathbb{R}^d$ outside of the discrete set of the rows of $\mathbf{E}$ have little meaning to the generative model $\mathcal{M}$. As a consequence, most prompt inversion methods focus on strategies for discrete optimization within the embedding table $\mathbf{E}$; we call this 'hard prompting' in a discrete space rather than 'soft prompting' in a continuous space. While the gradient exists with respect to the entries of the input $\mathbf{XE} \in \mathbb{R}^{s \times d}$, continuous descent-based methods risk finding minima outside of $\mathbb{T}^s$, leaving us without hard tokens.

Moreover, computing the gradient through the full generation model $\mathcal{M}$ may be too expensive (for example when $\mathcal{M}$ is a diffusion model forward passes may take multiple seconds), but as emphasized above, prior work often uses CLIP (Radford et al., 2021) to encode images and text in a shared latent space. Some of the methods we examine operate wholly within CLIP's latent space to compute the loss between the prompt and the target image. These methods approximate Equation (1) by solving the following problem where $\mathcal{L}_{\text{CLIP}}$ is a similarity loss defined over CLIP embeddings.

$$\operatorname*{argmin}_{\mathbf{X} \in \{0,1\}^{s \times |\mathbb{T}|}} \quad \mathcal{L}_{\text{CLIP}}(\mathbf{XE}, Y) \quad \text{s.t.} \quad \sum_{i=1}^{|\mathbb{T}|} X_{j,i} = \mathbf{1} \ \forall \ j \in \{1, ..., s\} \tag{2}$$

### 3.1   PEZ

The first approach we consider is PEZ (Wen et al., 2023), a version of projected gradient descent where descent steps are made in the continuous embedding space. The gradients of the objective in Equation (2) are evaluated at points in embedding space corresponding to real tokens, but the trajectory of the iterates may deviate from the discrete token set.

More formally, let $\texttt{Proj}_{\mathbf{E}}(\cdot)$ be an operator that projects vectors (or matrices row-wise) from $\mathbb{R}^d$ to their nearest row-vector of $\mathbf{E}$, and let $\xi_i \in \mathbb{R}^{s \times d}$ be a soft prompt. As an iterative

gradient-based optimizer, PEZ produces a sequence of iterates $[\xi_0, \xi_1, ..., \xi_n]$ as it solves the minimization problem in Equation (2). To update from $\xi_i$ to $\xi_{i+1}$, PEZ computes the gradient of the loss at the hard prompt $\text{Proj}_{\mathbf{E}}(\xi_i)$ and takes a step in the direction of this gradient from the soft prompt $\xi_i$ and then calls $\text{Proj}_{\mathbf{E}}(\xi_i)$ to project back to the space of hard prompts. Thus, PEZ gives a fast, lightweight method of discrete optimization while still using gradient-based descent to approximately solve the problem in Equation (1). For more information, see Algorithm 1 as described by Wen et al. (2023). For a single image, we run PEZ over the CLIP loss for 3000 steps and return the prompt the maximizes the CLIP similarity between the image embedding and the text embedding of the prompt.

## 3.2 GREEDY COORDINATE GRADIENTS

Greedy Coordinate Gradients (GCG) (Zou et al., 2023) is an alternative method for optimizing over the discrete vocabulary using the gradients of the objective with respect to the matrix $\mathbf{X}$ in Equation (2). In particular, we compute the gradient of the loss with respect to $\mathbf{X}$, which is a matrix of the same shape that approximately ranks token swaps. As each entry in a given row of $\mathbf{X}$ corresponds to a token in the vocabulary, each row $i$ in its gradient relays to us how influential changing the token $x_i$ to each other token in the vocabulary might be in lowering the loss. More formally, we compute $\nabla_{\mathbf{X}}\mathcal{L}_{\text{CLIP}}(\mathcal{M}(\mathbf{XE}), Y)$, then, just as gradient descent methods takes steps in the opposite direction of the gradient, we select a random batch of candidate swaps from the top $k$ largest entries of the *negative* gradient. A given swap corresponds to a single token change in $x$ and we directly compute the loss for each of these candidates and greedily accept the best one as our new iterate. As done with PEZ, we run GCG over the CLIP loss for 3000 steps, returning the best prompt as determined by CLIP similarity between the image embedding and the prompt's embedding.

## 3.3 AUTODAN

AutoDAN (Zhu et al., 2023) was proposed as a method of finding human-readable adversarial attacks on aligned language models. The optimizer solves Eq. (1) by iteratively optimizing a single token appended to the current prompt. Given an initial prefix, e.g., "Image of a", the algorithm searches for the token that follows 'a' that minimizes the objective function. The optimizer incorporates a 'readability' objective based on the log probability of the next token given an underlying language model. Similarly to GCG, AutoDAN employs a coarse-to-fine search strategy by appending an initial token, $\hat{x}$ to the current iterate $x$, and scores each token in the vocabulary according to the following scoring function:

$$\text{score}(x_i) = -\left(\nabla_{\hat{x}}\mathcal{L}([x, \hat{x}]E)\right) + \log(p(x_i|x)) \tag{3}$$

The algorithm selects the top $k$ scoring tokens and performs a fine-grained search by computing the exact loss over each, taking the token that minimizes the loss, $\mathcal{L}$. This minimizing token is then appended to $x$, giving $x_{t+1} = [x_t \quad x_i^*]$.

AutoDAN was originally designed for text-to-text language models, where the log probability, $\log(p(x_i|x))$ was directly available. However, in this review, we use CLIP to determine the quality of the prompt, which does not inherently compute the log probability. We thus use FUSE (Williams and Kolter, 2024), a recently proposed approach for solving multi-objective problems across models and tokenizers. FUSE approximates the jacobian of a mapping between the two models and uses the embeddings of a text-to-text language model, such as GPT2 to compute both the log probability, $\log(p(x_i|x))$, and the gradient, $\nabla_{x_{GPT}}\mathcal{L}_{CLIP}(f(x_{GPT}))$, where $f$ maps from GPT's embeddings to CLIP's embeddings. This allows us to apply a language prior when optimizing a prompt with CLIP. We additionally explore the scenario in which we do not use a language prior, by reverting to the standard case in which we fix $p(x_i|x) = \frac{1}{|\mathbb{T}|}$. In our experiments, we run AutoDAN for 16 steps, which enforces a a maximum token length of 16 due to one-by-one generation of new tokens. We also utilize a beam search with a beam width of 5.

## 3.4 RANDOM SEARCH

Andriushchenko (2023) suggests that such sophisticated strategies may not be critical for prompt optimization—given enough time, random searches can perform adequately in a

variety of settings. Thus, we explore a variant of random search (Rastrigin, 1963). While random search traditionally selects random candidates from within a ball around the the current iterate, this approach does not directly map to hard prompting. Due to the curse of dimensionality, true random samples around these high-dimensional embedding spaces are sampled from a ball of with negligible volume around the initial embedding; a nearest neighbor projection would often fail to return a new candidate.

In order to address this limitation, we randomly sample from new tokens from the $l_0$ ball around each element in the sequence **XE**. At every iteration, we select a batch of candidatesand greedily accept the best single-token replacement as the next iterate. We compare the prompt found by Random Searching over the same number of steps as done for PEZ and GCG, determining the best prompt by CLIP similarity in the same way.

### 3.5 PRISM

PRISM, proposed by He et al. (2024), highlights that text-to-image generation is not a one-to-one mapping – multiple prompts can describe the same image, and many images can correspond to the same prompt. Rather than relying on discrete token space optimization, PRISM optimizes over a distribution of prompts. Inspired by LLM jailbreaking methods (eg. Chao et al., 2023), PRISM leverages in-context learning in vision-language models (VLMs) to iteratively refine the prompt distribution. This process incorporates the history of reference images, generated prompts, output images from an anchor text-to-image model, and evaluations from a VLM judge, using techniques similar to chain-of-thought (Wei et al., 2022) and textual gradient (Pryzant et al., 2023). After $K$ iterations across $N$ parallel streams, the best-performing prompt is selected using the same VLM judge. In our experiments, we use GPT-4-o-mini as the VLM and Stable Diffusion XL-Turbo (Sauer et al., 2023) as the anchor text-to-image model, following He et al. (2024)'s setup with $N = 6$ and $K = 5$. To ensure fair comparisons, we limit the generated prompts to 20 tokens.

### 3.6 CAPTIONING

Lastly, we use automated image captions as a proxy for the inverted prompts. Given that a prompt for an image generation model likely encodes information about the setting of the desired image, its subject, its quality, and other properties, we assume that captioning an image provides a human-readable token sequence with some or all of these same properties necessary to generate the image. Moreover, as captioners are typically autoregressive, they have the potential to return an approximate inversion much faster than other methods.

Here, we focus on a single model, BLIP-2 (Li et al., 2023). This model is a generic and compute-efficient vision-langauge pre-training (VLP) method. VLP techniques aim to learn multimodal foundation models on a variety of vision-language tasks. BLIP-2 leverages a trainable module, the Q-former, in order to bridge the gap between a frozen image encoder and a frozen LLM, facilitating image-text matching tasks, image-grounded text generation tasks, and image-text contrastive learning tasks. We prioritize BLIP-2's image-grounded text generation as the frozen CLIP-style encoder aligns well with the above prompt inversion methods, all of which use frozen CLIP encoders.

## 4 EVALUATION

For each optimizer detailed above, we assess their performance across several criteria. Considering the stochastic nature of image generation, we measure the effectiveness of an inverted prompt by asking the following questions.

1. How similar (FID (Heusel et al., 2017), KID (Bińkowski et al., 2018)) are images generated with the inverted prompt to images generated by the original prompt?

2. How well (CLIP (Hessel et al., 2021)) do the inverted prompt and original image align?

3. How well (Text Embedding Similarity (Reade et al., 2023)) does the semantic content of the inverted prompt align with the semantic content of the original prompt?

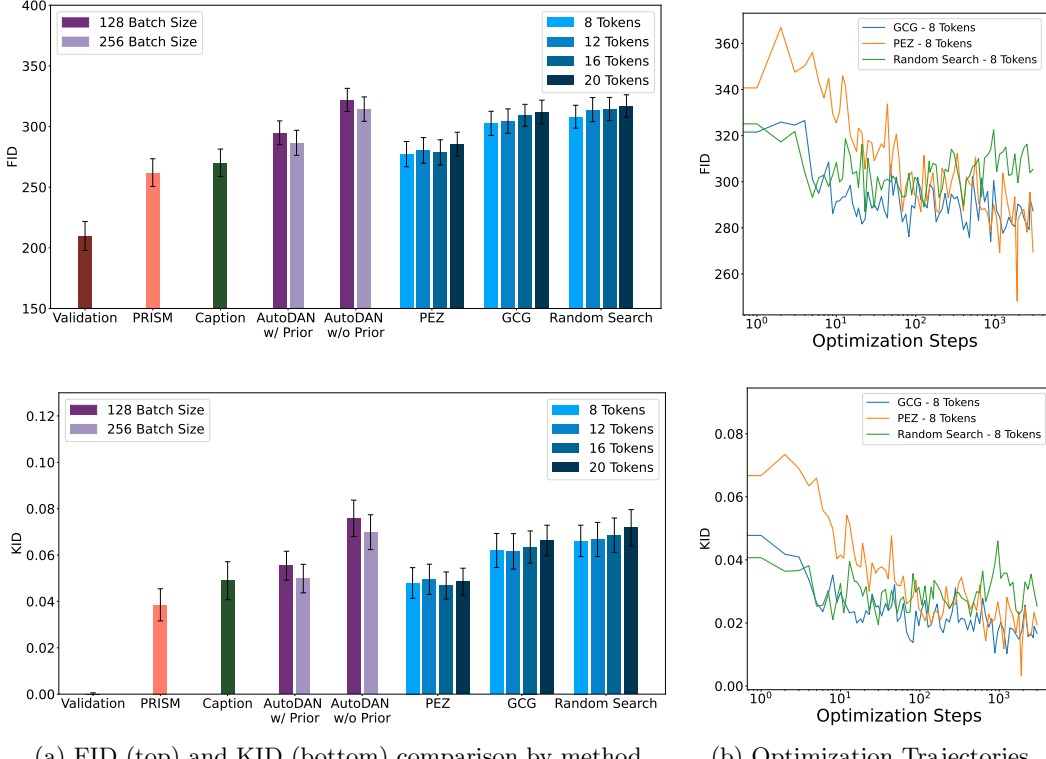

(a) FID (top) and KID (bottom) comparison by method    (b) Optimization Trajectories

Figure 1: Comparison between images generated by inverted prompts and images generated by the original prompts.

We address the stochasticity inherent to the image generation process by averaging the performance of each method across several images generated by the original prompt and the inverted prompts. First, we randomly sample 100 prompts from an existing dataset of prompts[1] used by Stable Diffusion (Rombach et al., 2021).[2] Given each of these prompts, we generate 10 baseline images for each baseline prompt, and invert each according for all of the 7 methods considered here. Once we have found an inverted prompt for each baseline image, we generate 2 images for each inverted prompt, and finally compute our metrics across the 10 baseline prompts and images and the 20 images based on the 10 inverted prompts. In addition, we choose 75 log scaled time points within the 3000 optimization steps used for PEZ, GCG, and Random Search and repeat our full analysis on a subset of DiffusionDB prompts in order to better understand the convergence of each method.

## 5 EMPIRICAL RESULTS

In this section we present quantitative and qualitative results comparing each method. Across several metrics, we see the quantitative rankings are consistent, but we find upon qualitative examination that these numeric rankings show only a partial picture. Examining the images and the recovered prompts themselves we see trade offs across methods.

### 5.1 QUANTITATIVELY RANKING METHODS

**Image to Image Comparisons**   For image to image comparisons (Figure 1), we analyze images generated by the best early-stopped prompt for each method and the convergence

---

[1]We use samples from the Poloclub DiffusionDB dataset of prompt-image pairs (Wang et al., 2022) to find our evaluation prompts.

[2]All images are generated with StableDiffusion 2-1: `stabilityai/stable-diffusion`.

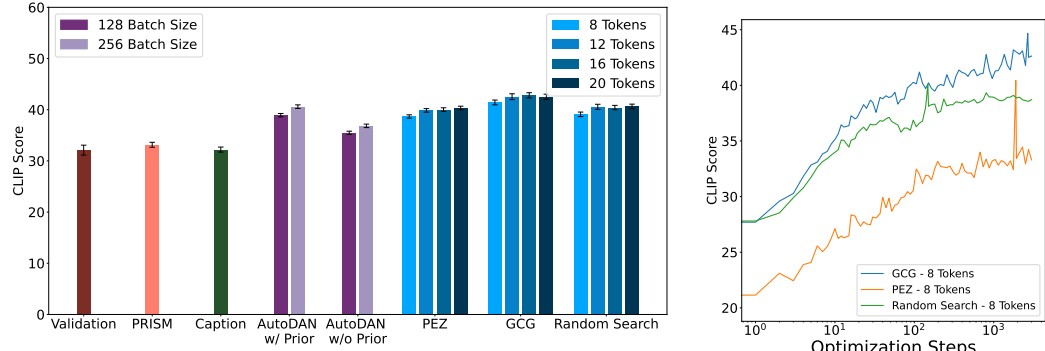

Figure 2: CLIP Similarity between the inverted prompt and images generated by the original prompt. This CLIP Similarity is the objective that each optimizer is maximizing.

rates across our considered image similarity metrics for each algorithm. Our validation set, which consists of the ground truth prompts has an FID of 209.78 and a KID of −0.001. The KID score in particular tells us that the closer any algorithm gets to a KID of 0, the more similar that prompt will be to the ground truth, whereas, while the ordering may be consistent with FID scores, it is possible that using FID rather than KID may incorrectly show that a method improves over the validation set.

We find that generating images from PRISM prompts provide the most similar images to those generated by the original prompt, with those images generated by BLIP-2 and PEZ as close seconds; PRISM gives average FID and KID values of 262.015 and 0.0385 respectively, while the captioner generates images with average FID and KID values of 270.085 and 0.0489. PEZ follows these with average FID and KID values of and 280.392 and 0.0482. In addition, we see a significant gap in performance between AutoDAN with a prior and AutoDAN without a prior, where the former performs much more similarly to the captioners and the latter performing in line with GCG and a Random Search.

Analyzing the objective trajectory over the course of optimization reveals interesting trade-offs. We used a small validation set to determine the number of steps for all algorithms to converge for the given prompts and images used in this study. We determined that all optimizers stop receiving meaningful improvements after 3000 steps. We observe that GCG and a Random Search find a prompt comparable to their best early-stopped prompt within the first 25 steps and then struggle to descend further, analogous to applying too high of a learning rate to optimization problems. On the other hand, PEZ has a slower convergence, but it descends consistently across all steps until it finds prompts that improve over both the GCG and Random Search prompts. Moreover, as PEZ uses a single forward and backward pass, it requires much less time to run than the comparison methods. In other words, PEZ finds prompts that generate images more similar to the ground truth in much less time than all other optimizers considered here, except for the BLIP-2 Captioner.

**Text to Image Comparisons**  When we focus on the alignment between the text and images we see an interesting trend emerge. We first compare the CLIP similarity between the inverted prompts and the original image (in the top of Figure 2). Note that this is the optimization objective used across all optimizers.

We find that all optimizers do a good job maximizing their objective. While AutoDAN without a language prior performs the worst over all optimizers, it still does a better job of maximizing the CLIP similarity over the validation set, PRISM, and the BLIP2 Captioner. Optimizing the objective with GCG and AutoDAN with a language prior performs the best over the discrete optimizers, with PEZ coming a close third. The contrast between the performance of each optimizer on their objective and their relative lack of performance across the image-to-image and text-to-text metrics suggests that the CLIP objective is acting as a poor proxy for finding prompts for generative image models. While there may be room for improvement over the CLIP objective for this task, this comparison allows us to take

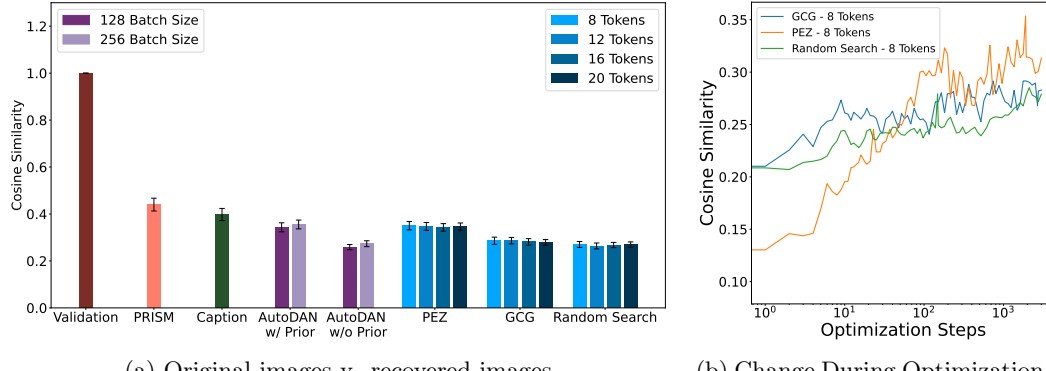

(a) Original images v. recovered images

(b) Change During Optimization

Figure 3: Cosine Similarity between text embeddings for the original and inverted prompts. Based on the metric used by Reade et al. (2023)

a better look at the convergence rates of all optimizers on their objective. Just as in the image-to-image comparison, GCG and Random Search quickly find a good prompt (within 20 steps) and then very slowly improve from there.

Yet PEZ follows a much more gradual curve, with sharp peaks when new optima are found. As these are log scaled in their x-axes, we do not see all peaks except for the early stopped result. The average prompt found with PEZ is much lower than the comparison methods, but the peaks are in line with the other methods. Additionally, GCG and Random Search again very quickly within the first 20 steps and then very slowly update from there. This overreliance on early-stopping may be a weakness for PEZ. Rather than oscillating tightly around the optima, PEZ oscillates wildly around high quality prompts. In essence, PEZ may better explore the prompt space, while methods incorporating fine-grained search (such as GCG) are more adept at exploiting it.

**Text to Text Comparisons** Lastly, we compare the similarity in the text of the found prompts to the ground truth prompts. Figure 3 shows the cosine similarity between the text embeddings[3] of the found prompts and the ground truth prompts.

Just as in the image-to-image case, we find that using responses from PRISM as the inverted prompt outperforms all of our comparisons, with a cosine similarity of 0.440 to the original prompt; the BLIP-2 captioner comes in second with a cosine similarity of 0.397 to the original prompt. AutoDAN with a language prior and PEZ follow behind with respective similarities of 0.355 and 0.346 averaged across all lengths. GCG, Random Search, and AutoDAN without a language prior remain clustered together in terms of their performance. Moreover, when looking at their convergence rate, we see the same story as above. GCG and Random Search very quickly ascend, and while PEZ ascends more slowly it eventually exceeds GCG and Random Search in their performance within the first 100 optimizations steps of its allowed 3000 steps.

**Hyperparameter Choice** Each optimizer has various hyperparameters that can be adjusted. Intuitively, it may seem that the number of free tokens (tokens that can be optimized) is a particularly relevant parameter. However, this is not always the case. In line with the findings by Wen et al. (2023), PEZ's performance remains fairly independent of the number of free tokens up to a certain point. Given that we allowed no fewer than 8 free tokens, there may be a performance drop-off if we further decrease the number of tokens. However, PEZ's performance across all metrics does not appear to be significantly dependent on its free tokens. For context, the BLIP-2 captioner, which does not have a fixed length, can serve as a benchmark for reasonable prompt length. Its captions average 10.6 tokens using CLIP's tokenizer, and 5.8 tokens after removing stop-words. Similarly, we see that neither

---

[3]Embeddings were computed using **sentence-transformers/all-MiniLM-L6-v2** to be in line with (Reade et al., 2023)

| | |
|---|---|
| **Original Prompt** | a friendly goblin with a big ( ( human ) ) nose and wide eyes, dark hairs, big ears, covered in branches and moss, portrait by daniel docciu and dave dorman and jeff easley |
| **PRISM** | Friendly green goblin face, smiling, tangled branches, soft forest background |
| **BLIP-2** | green troll in a tree with leaves and branches on it's head and a smiley face on it's face |
| **AutoDAN w/ Prior** | character from Magic Treefolk depicting Green Elf head with smile during 2015 promotional image walkthrough art group image |
| **AutoDAN w/o Prior** | animated jester troll grass wordpress goblin frightening branch artwork today )) oman ly 6 jester head. _ reid / |
| **PEZ** | newmlb mtgnflrevealed loki revealed reveal ).. goofy smiling scary creepy ytree ï orc =)) arbormates |
| **GCG** | typically greener donny recent reported atrist......... frightening cohen substantially _: . kal ears googmirrowickedcriticalrole trees believes |
| **Random Search** | cantenzenegger oaks ]m̃odo grin grassy goblin ytless...( ĩ ¿·¡ sends iconforeveryimp huskdns |

Table 1: Example images and corresponding 20-token prompts. Each image is generated by the *original prompt* and we show examples of the inversion result from each method. Other AutoDAN with the language prior applied, no discrete optimizer produces more human-readable prompts than another despite the quantitative differences their performance.

GCG nor Random Search dependent on the number of free tokens. With no metric showing a statistically significant improvement for when adding more or fewer optimizable tokens.

AutoDAN, like the BLIP-2 captioner, can return a variable number of tokens due to early stopping. However, we use AutoDAN with a beam search, where each individual step uses a much smaller batch size for its fine-grained search, unlike GCG and Random Search. The latter have a 512 batch size, while AutoDAN with 4 beams and 128 tokens evaluates the same number of tokens for the fine-grained search. Increasing AutoDAN to 256 tokens, effectively doubling the number of tokens it searches over compared to GCG and Random Search, results in a small improvement across the board. Based on the optimal batch sizes described by Zou et al. (2023), further improvements in AutoDAN might be achieved by allowing a 512 batch size. However, there are likely diminishing returns, especially as computation time increases with larger batch sizes.

## 5.2 Qualitatively Assessing Inverted Prompts

In the quantitative evaluation above, we show that PRISM and the captioner return prompts that may be better across several metrics compared to searching for a prompt via discrete optimization. Here, we show a qualitative example (Table 1) of an image generated by one of the ground truth prompts and the different results that each method find. Other than AutoDAN with the language prior applied, no discrete optimizer produces human-readable prompts despite the quantitative similarities in their performance. We thus separate this subsection into natural language and keyword-based prompts that without a language prior.

PRISM provides prompts that are exceptionally more detailed than the comparison methods; opting for short descriptive clauses rather than the full sentences that BLIP-2 uses. As described above, when the length of a prompt is limited, the additional stop words required by full-sentence prompts reduce the number of concepts that can be included in a prompt,

significantly affecting the final image. On the other hand, AutoDAN's language prior seems to finds natural language prompts that evoke the imagery described by the image, "character from Magic Treefolk..." does not describe anything from the image (to the best of our knowledge there is not media called Magic Treefolk), but if such media existed then we would not be surprised to find that something called Magic Treefolk included depictions of goblins or other forest critters.

When comparing each recovered prompt to the original prompt, there is often a significant amount of information lost during the generation process that is unrecoverable. Both Random Search and PEZ capture basic information such as "trees" or "green". These methods try to included the single tokens that encode as much information as possible. Similarly to the "Magic Treefolk" example above, GCG uses the token "criticalrole" for a similar purpose. Critical RoleMercer (2015–present), a 'Dungeons & Dragons'-based web series, embeds a relationship between the prompt and creatures found in Dungeons & Dragons through a single token. Moreover, without the need for a language prior, it does not need to waste tokens fitting 'criticalrole' into a coherent sentence. Yet, it may cause an overreliance on these 'keyword' tokens and allow unrelated tokens such as 'goog' to be included in a prospective prompt. This comparison may shed light on why PEZ outperforms GCG and random search, as PEZ appears to stay more on topic. PEZ includes the tokens "loki", "tree", "arbor', "scary", "goofy", "orc" and "smiling", while GCG and Random Search do not provide significantly more specificity than "green", "trees", and "criticalrole"; and "oaks", 'goblin" and "grassy" respectively. At its core, PEZ is a projected gradient descent method, using common optimizers, such as SGD or Adam with a weight decay. This approach encourages some form of regularization in its optimization, that discourages the one-and-done approach that GCG and Random Search seem to use, where they discourage repeating the same general concepts or tokens in a prompt.

## 6 DISCUSSION

Our results prompt discussion on the practical implications, the limitations, and the future directions related to prompt inversion. To begin, someone interested in finding good prompts from images can conclude from our work that image captioning tools are a good approach. They are fast, as the heavy lifting is done ahead of time in training these models rather than optimizing anything per image in deployment. They also best capture natural sounding language, a goal that discrete optimizers might better incorporate as these tools mature.

The limitations of our work center mostly on the fact that the diffusion and image-text embedding space is so heavily driven by only a few models. As the set of state-of-the-art large text-prompted image generations models grows, the trends we report may no longer hold. In the same vein, small variations in the optimization strategies could have large impacts on these results. In short, like any empirical benchmark results, our findings are subject to change as the field progresses.

Here, we enumerate several questions and quirks arising from our work that warrant further investigation. First, Zou et al. (2023) report that GCG is effective at jailbreaking LLMs and PEZ is not. This stands in stark contrast to these two methods relative performance at prompt inversion. Why might optimizing over natural language be so different in these settings? This could be a difference in the particular models or in the loss landscapes. Second, GCG and random search perform so similarly begging the question why does gradient information make so little difference? The intuition that the gradient signal is informative comes from observing the success of PEZ, so why is the combination of search and gradient-based optimization in GCG leave it so similar to random search alone? Finally, we posit that there is a lot of room for improvement. In other words, prompt inversion is far from solved, and it makes for a great test bed for new discrete optimization approaches.

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
