# OpenReview forum: "Prompt Recovery for Image Generation Models: A Comparative Study of Discrete Optimizers"
_ICLR.cc/2025/Conference — Submitted to ICLR 2025_

### Official Review · Reviewer_4nSo · 2024-10-19

**Soundness:** 1
**Presentation:** 3
**Contribution:** 1
**Rating:** 3
**Confidence:** 4

**Summary:**

The paper investigates solutions for the task of prompt inversion, where the goal is to recover discrete text-to-image prompts that lead to the reconstruction of images from the same distribution as a given target image.

Specifically, they consider a set of existing discrete optimization techniques, as well as a feed-forward captioning model and an in-context learning based approach that leverages a VLM judge to iteratively refine prompts.

To evaluate these, the authors create a set of images with a fixed prompt benchmark, invert the prompts using each method, and then generate novel images using the inverted prompts. These new images are evaluated against the original generated images using standard quality metrics (FID, KID). Furthermore, CLIP-score is used to compare the new images to the original prompt set, and a text-to-text CLIP similarity metric is used to evaluate the inverted prompts.

The authors conclude that CLIP similarity between inverted prompts and the ground truth images is a poor proxy for the prompt inversion task, and that further research is needed in this direction.

**Strengths:**

The paper’s main strength is the attempt at a more thorough evaluation and better benchmarking of methods in the prompt-inversion field. The metrics used are mostly reasonable, and are used to evaluate a range of baselines, which are clearly presented in the paper.

Overall, the paper was also fairly easy to read, the experiments easy to understand, and I believe this paper could be fairly easily replicated.

**Weaknesses:**

Unfortunately, the paper is very limited in scope and grounds some of its choices in claims that are probably incorrect. For example:

1)
The motivation for focusing on discrete prompts (L138) is probably wrong.
The findings of Khashabi do not seem relevant in this scenario since the text-to-image field uses soft prompting very extensively (Textual Inversion [Gal et al, ICLR 2023], Null-Text Inversion [Mokady et al, CVPR 2023], Custom Diffusion [Kumari et al, CVPR 2023], Imagic [Kawar et al, CVPR 2023], Reversion [Huang et al, 2023]).

Similarly, the claim “As a consequence, most prompt inversion methods focus on strategies for discrete optimization within the embedding table E” is likely wrong. This is difficult to gauge, but prominent soft prompting techniques in the text-to-image setting are cited 10-20 times as often as their discrete counterparts.

2)
The work focuses on CLIP-based inversion, where CLIP-space similarity is used to gauge the alignment between prompts and images. This is a very limiting restriction, and likely leads to much of the author’s conclusions.

A lot of prior art does use the diffusion model to compute a loss, and the signal there is stronger and more aligned with the diffusion distribution itself (see e.g., all the papers cited in point (1) above, or the work of Mahajan et al (CVPR 2024) which outperforms PEZ across the board.

If a range of methods is being benchmarked, it would be revealing to see how different families of approaches behave when compared with each-other, and specifically if and how the use of a diffusion loss changes the findings on CLIP-based optimizers.
This holds doubly true these days since most large-scale models are shifting away from using CLIP as the text encoder and its not clear if the findings here will even generalize to such models.

3)
The authors demonstrate that CLIP alignment is decoupled from FID, KID improvements and conclude that CLIP is a poor proxy for the prompt inversion task (L374). However, most of the benchmarked methods use the same CLIP model for a loss and for the evaluation. I am not entirely convinced that this is this a matter of CLIP acting as a poor proxy, instead of the proposed methods simply finding solutions which are adversarial to CLIP (see e.g., EditBench [Wang et al, CVPR 2023] which arrive at an opposite conclusion regarding CLIP). When you consider AutoDAN with the prior, for example, you get improved results on the visual side, indicating that making it harder for the model to find adversarial solutions may help?

It would be informative to see CLIP results also using a different CLIP model (e.g., OpenCLIP). This could help somewhat in disambiguating adversarialness vs a disconnect between CLIP and image alignment.

Additional concerns:

4)
The evaluation is relatively basic. In a paper focused on this kind of benchmarking, I would have expected to see far more in-depth experiments. For example, does the performance of different approaches depend on the type of image being inverted (style reference, dense scene with lots of content, single vs multiple subjects etc.)? Do these findings generalize across different generative models? What about models not using CLIP as a text encoder?

5)
I would have liked to see human evaluation to back up most of the core experiments in the paper (even at a smaller scale).

6)
There is only a single qualitative example of inverted prompts, and no examples of the reconstructed images for any of the methods. Please add some visual examples.

I will note that my biggest concern is the extremely limited scope of the work, and I’m not sure this can be addressed in a rebuttal. However, I’m willing to discuss matters if the authors believe they can convince me otherwise.

**Questions:**

See questions and suggestions in the weakness section.

Particularly, if the scope of the work remains just CLIP-based discrete inversion methods, I would have liked to see more extensive evaluation and insights into the specific scenarios where different methods excel or underperform. Moreover, I would have liked to see a more robust evaluation of the claims surrounding CLIP being a poor proxy (trying to control for adversarial solutions, adding human evaluations).

---

> ### Author Response · Authors · 2024-11-14
>
> Thank you very much for your review. We agree that there are several areas for improvement, and you have highlighted important considerations for future revisions. Here, we address what appear to be your primary points of disagreement with our work.
>
> We see value in studying hard prompts, despite their greater complexity compared to soft prompts, particularly for applications in explainability and safety. Our analysis and results are intended for researchers interested in hard prompts as an alternative to soft prompting. You are correct that soft prompting is more prevalent in the literature, and in future revisions, we will rephrase statements like “most prompt inversion methods focus on strategies for discrete optimization within the embedding table E” to more clearly reflect our hard-prompting perspective. However, we disagree that our motivation is flawed. Our reference to Khashabi et al. is not intended to invalidate soft prompting. Their work demonstrates that, for a fixed hard token and classifier, soft tokens whose nearest neighbor is the fixed hard token can yield arbitrary classifier outputs. We interpret this to mean that, while soft prompts can produce accurate images, they are not necessarily useful for assessing the fitness of a given hard prompt; this does not imply that soft prompts are inherently meaningless. We will adjust language here in future revisions.
>
> Regarding the limitations of CLIP similarity, while we acknowledge this as a constraint, we do not believe that alternate versions of CLIP or other VIT models would alter our conclusions. This paper is focused on analyzing discrete optimization methods rather than exploring CLIP’s suitability as a proxy for Stable Diffusion. Our findings show that, while the captioner and PRISM outperform the optimized prompts, our primary conclusions are based on the discrete optimizers’ performance and their behavior.
>
> Finally, we do not believe our results contradict EditBench. EditBench demonstrates that CLIP aligns with human evaluations of prompt relevance; it does not claim that CLIP performs on par with humans in prompt construction. Our results indicate that discrete optimizers can identify adversarial prompts, and without stronger proxy models or improved exploration of the hard token space, they are unlikely to surpass captioners—and, by extension, human annotators—in prompt generation.

---

> > ### Comment · Reviewer_4nSo · 2024-11-21
> >
> > Thank you for your response.
> >
> > I appreciate that an effort into analyzing the behaviors of different hard-prompting approaches could be valuable, even when such methods are not the dominant tool in the field. Hard prompting could certainly have benefits like better transferability between models, better interpretability and more. My note was regarding the presented motivation for focusing on them, rather than on the question of whether hard-prompting is valuable.
> >
> > With that said, when I ask myself if the findings of a paper are of interest to the community, I consider whether they: (1) can be directly applicable in some way that would influence ongoing or future research, (2) provide some novel understanding about the tools or methods that are being analyzed, or (3) introduce some interesting new idea or concept.
> >
> > Regarding point (1), this is where my concern with limiting results to just the CLIP-based scenario arises from. Since recent text-to-image models are moving away from CLIP encoders, and since there already exist hard-prompting approaches that do not suffer from the CLIP-based restriction, then limiting yourselves to CLIP-only approaches likely means that future research will have a harder time building on your results. It may be that hard tokens can generalize well across text-encoders, but this needs to be supported empirically.
> >
> > On point (2), I do not think there is much novelty in showing that optimizing for a specific loss can lead to solutions that are adversarial to that loss (I could fetch some citations going back to Goodfellow et al., but I believe this can be filed under 'well known'). From reading through the paper, I was also under the impression that the claim is that CLIP is a poor metric for evaluating hard prompting techniques, and not that the specific set of CLIP-loss based techniques should not use it because its adversarial. This is also where the EditBench results come in - showing that CLIP is indeed a good metric for prompt alignment in the general case, which raises the worry that non-CLIP optimized hard-prompting techniques will not show the same disconnect between FID and CLIP scores. A paper showing that CLIP remains a poor, un-aligned metric for hard prompting regardless of the optimization loss would be much more interesting since this would be a very surprising finding.
> >
> > On (3) - This is an analysis paper using standard tools so this is not really applicable.
> >
> > Overall, I am inclined to keep my current rating for this version of the manuscript. If you think it would be valuable for future submissions, I'm happy to discuss things further.

---

### Official Review · Reviewer_Uzm8 · 2024-11-01

**Soundness:** 2
**Presentation:** 2
**Contribution:** 1
**Rating:** 3
**Confidence:** 4

**Summary:**

The paper presents an empirical study of different techniques for prompt recovery in text-to-image diffusion model. The idea is to select 6 baseline methods (PRISM, Captioning, AutoDAN, PEZ, GCG) and assess their performance in recovering the input prompt for an input image. The study is conducted by sampling 10 images for each one of 100 prompts used as input in Stable Diffusion 2.1, where the prompt is reconstructed by each of the proposed methods. The quality of the reconstruction is then evaluated with FID/KID score among images of the original and reconstructed prompts, CLIP score among inverted prompts and original images, and text embedding similarity between the original and inverted prompts.

**Strengths:**

I recognize the novelty of the proposed approach. To the best of my knowledge, this is the first work comparing different prompt inversion techniques for text-to-image models. The paper is overall well-written and easy to follow and understand.

**Weaknesses:**

I have several concerns about this manuscript. Overall, I believe that unfortunately, the proposed study lacks the importance of the results to be at the level of ICLR. I will detail my viewpoint in separate sections.
1. My main concern is related to the impact of the proposed results on the community. In a comparative study like this, I believe the most important factor is providing the community with a new viewpoint on an important issue, potentially encouraging new research or raising new questions. I struggle to see this in the proposed manuscript. By reading the paper, it looks to me that results are just presented, without further interpretation that may give new perspectives. There is a Discussion section but it is quite rudimentary, stating that captioning is a good alternative to prompt inversion and re-assessing questions already known in the literature, properly cited.
2. I also believe that the experimental effort in this study is quite limited. There is only one diffusion model considered (SD v2.1), 1000 generated images for each method under analysis. This can be done in just a few days in any laboratory with sufficient computational power. The evaluation itself is quite limited. For instance, there is no comparison of the effect on fine-grained details of prompt inversion with respect to captioning, nor a proper analysis of the variability of results for the same prompt (variability is included in the experiments but not treated separately). For the qualitative evaluation, there is no user study allowing to trace of new conclusions following human judgment.
3. While the text is easy to follow, I believe that Section 3 of the paper can be completely removed and replaced with a paragraph, to allow for more space for experiments, which is highly needed in this case. Why describe all methods in detail when we can simply refer to the literature and quickly summarise them? The entire section feels unnecessary.

**Questions:**

1. What new observation we can have, as a community, on different prompt inversion techniques, and why this is important?
2. How can we complement this analysis with proper results on multiple aspects of prompt inversion?
3. Why Section 3 is needed rather than other experiments?

---

> ### Author Response · Authors · 2024-11-14
>
> Thank you very much for your thoughtful review. We agree that there are several areas in which this work can be improved, and we appreciate your insights on potential directions for future revisions. We agree that the discussion could be expanded to better articulate the main points of potential impact for the community.
>
> A key clarification, both here and in future revisions, is that our focus is not on identifying which method generates the best prompt for image creation. Rather, we aim to compare discrete optimization approaches for images, highlighting the benefits and potential concerns of each method. We use captioners as validation tools to assess whether and how discrete optimizers perform relative to captioners trained on human annotations. While our use of CLIP as a proxy is a limitation, improved proxies or methods differentiating directly through the diffusion model might affect prompt quality outcomes, but we do not expect them to alter the comparative advantages and drawbacks of each optimization method (see our response to reviewer uWcM for more on excluding methods that differentiate through the diffusion model).
>
> We believe the qualitative evaluation section offers valuable insights for future work in discrete optimization. Your suggestion on Section 3 is helpful, and we agree that subsections 3.5, 3.6, and possibly 3.4 could be considered for removal. However, we detail each method in Section 3 to provide a taxonomy of approaches to discrete optimization: (3.1) Projected Descent, (3.2) Coarse-to-Fine Search, (3.3) Fluency-based Searches, and (3.4) Purely Random. Our discussion in Section 5.2 about PEZ’s regularization of prompts, for instance, offers insights into the strengths of projected descent methods, while GCG’s tendency to find adversarial prompts highlights properties of Coarse-to-Fine Search methods that could inform improvements. We recognize that the current structure might obscure these points and agree that expanding our discussion would be beneficial.

---

> > ### Comment · Reviewer_Uzm8 · 2024-11-29
> >
> > I thank the authors for the additional clarifications. While I acknowledge that the proposed direction could be interesting, it is also true that such paper should be associated to novel findings, that should bring some new insight and directions for future research to build on top. In the current state of the paper, it looks to me that these findings are very limited, and as such I do not think that the manuscript is of sufficient depth to be accepted to ICLR. I will keep my score unchanged, but I encourage authors to further refine the research and aim for more substantial findings to bring their submission to an A* venue level.

---

### Official Review · Reviewer_uWcM · 2024-11-04

**Soundness:** 2
**Presentation:** 3
**Contribution:** 2
**Rating:** 3
**Confidence:** 5

**Summary:**

This work performs a comparative study on the different discrete optimization frameworks for hard prompt optimization. The methods compared are PEZ, PRISM, GCD, random search and captioning based models. The paper performs qualitative and quantitative analysis on different discrete optimization strategies for finding the best prompt for text-to-image generation. Experiments are performed on a set of 100 prompts from Stable diffusion and the inverted prompts along with the generated images are evaluated on the CLIP-scores, Cosine-similarity, FiD and KiD metrics.

**Strengths:**

+ The paper is easy to read and self-contained.
+ The work performs a comparative analysis on different prompt optimization strategies for text-to-image generation across different metrics such as CLIP-scores, FiD, and KID.

**Weaknesses:**

- The work considers Stable-diffusion 2.1 which uses a fixed pretrained CLIP-based text encoder. Therefore, the study will yield biased results towards approaches which minimize CLIP objective or have same distribution of image-prompt pairs. For example, PEZ optimizes in the CLIP space and therefore has better CLIP-scores in quantitative analysis.

- The work concludes captioning based models to yield better prompts for generating images. However, this is not necessarily true. The vocabulary and the distribution of a captioning model and the image generation model can be very different.

- Prompt inversion directly from SD model as in Mahajan et al. 2024 is not considered. The diffusion objective with a PEZ -like optimizer would provide insights into the impact of loss in eq.1 and eq.2 of the paper.

- For text-to-image generation, is the fluency of the prompt an important factor?

Minor:
The formatting is different from the ICLR template (package times in latex).

**Questions:**

- Are the results biased towards models which actually perform inversion (not captioning based) in the CLIP-space. Can an independent model be used to verify the validity of the claims?

- As mentioned in the weaknesses, for the claim that captioning based models are better suited, the vocabulary of the pretrained text-to-image diffusion model should align with that of the captioning model?

---

> ### Author Response · Authors · 2024-11-14
>
> Thank you very much for your thoughtful feedback. Regarding the study's limitations, as you noted, we focused on Stable Diffusion specifically for its CLIP-based text encoder. Our primary goal is to compare discrete optimization methods—PEZ, GCG, AutoDAN, and a baseline Random Search—validated against BLIP-2 and PRISM. We intentionally used CLIP due to its alignment with the image generator, providing what we believed to be the best available proxy for the generative model. However, our results show that while the optimized prompts do meet their objectives, they are outperformed on other metrics, suggesting that CLIP may indeed be an insufficient proxy. Nonetheless, our investigation is focused more on the behavior of the discrete optimizers than on prompt quality. We do not expect each optimizer’s performance to be dependent on the model or text encoder used, and we will clarify this point in future revisions.
>
> We agree that a comparison to Mahajan et al. would have been beneficial. Unfortunately, at the time of writing, no public implementation was available, and the authors did not respond to our requests. To ensure that any future comparisons are accurate, we opted to exclude this method for now and instead used CLIP to align with PEZ results [Wen et al., 2023] and Stable Diffusion’s text encoder.
>
> Regarding vocabulary and token distribution in the captioning method, we consider this issue minimal, as tokenization is inherently non-invertible. With hard token optimization, a set of 16 tokens optimized with the same text encoder may retokenize to 15 or 17 tokens. Consequently, any differences in vocabulary between captioners should not result in an advantage for one method over another when retokenized.

---

> > ### Comment · Reviewer_uWcM · 2024-11-30
> >
> > Thank you for your response.
> >
> > I think the paper in its current form does not provide any concrete findings, for example, it is limited to the CLI P-based encoders.   The authors argue that their findings are agnostic to the choice of text encoder however, the empirical evidence for this is not provided.
> > Regarding the difference in the vocabulary of the captioning and t2i generation approach, what I meant is that they are trained with different objectives, while t2i approaches focus on semantics, the captioning methods are optimized for fluency, therefore, the objective and the method can have an impact on the generation quality with recovered prompts.
> >
> > Given that the findings are limited, I will keep my score.

---

### Official Review · Reviewer_BePT · 2024-11-04

**Soundness:** 3
**Presentation:** 4
**Contribution:** 2
**Rating:** 3
**Confidence:** 3

**Summary:**

This paper presents an analysis of existing prompt optimization methodologies, including PEZ, Greedy Coordinate Gradients, AutoDAN, Random Search, PRISM, and Captioning. It contains extensive quantitative analysis of 100 prompts and images from the Stable Diffusion model across image:image, text:image, and text:text metrics for overall perfrmance and performance throughout the optimization process (for relevant methods). In addition, it includes a discussion of a qualitative prompt example and possible concept relations between specific terms in the prompts and original image.

**Strengths:**

* The paper is well-written, clear and easy to read with methodologies explained well and experimental set-up very clear
* The analysis contains broad coverage over six different prompt recovery methodologies
* The provided qualitative analysis is very thorough for the given example, usefully highlighting connections between related concepts mentioned in prompts and rarity/imagniation of certain terminologies.
* The quantitative analysis covers a variety of evaluation types, including image:image, text:image, and text:text, including between original and updated images, new prompt and original image, and new prompt and original prompt.
* The work is well-motivated and highly contextualized among previous works, defining why prompt inversion techniques are useful compared to other methods.
* The work usefully adapts existing methods used in language models to allow extensibility for text-to-image generative models.

**Weaknesses:**

1. While the provided analysis is insightful and sets the groundwork for improved methods, the work lacks thorough discussion or early experimentation of proposed solutions to the challenges observed with the existing prompt inversion methods. The contribution of the work is limited without studies or even specific proposals for improvements of existing methods contextualized by findings of the paper.
2. CLIP has known issues as a consistency metric, including bias in its representations [1,2] and issues with compositionality [3,4], that could make CLIPScore vulnerable. More robust alternatives such as DSG and VQAScore should be considered for the text:image analyses.
3. The analysis is only used on images generated from Stable Diffusion, thus limiting its insights. The work would be strengthened with study of prompt inversion techniques for images that come from other models, e.g. DALLE.
4. The qualitative analysis, while very thorough and insightful, seems limited to only a single image. This limits its utility and the takeaways may not be extensible to other prompt styles or concepts.

[References]
1. Evaluating clip: towards characterization of broader capabilities and downstream implications
2. Vision-Language Models Performing Zero-Shot Tasks Exhibit Disparities Between Gender Groups
3. Winoground: Probing Vision and Language Models for Visio-Linguistic Compositionality
4. WHEN AND WHY VISION-LANGUAGE MODELS BEHAVE LIKE BAGS-OF-WORDS, AND WHAT TO DO ABOUT IT?

**Questions:**

[Questions numbered according to each Weakness above]
1. The Discussion includes "details and quirks" for further investigation. Could there be more clarity about possible experimental directions to take steps towards improving the methods informed by these insights? Are there preliminary studies enabled by the existing analytical set-up that can provide initial guidance into these direction?
2. Do quantitative analyses using CLIPScore for text:image analyses hold when extended to more robust consistency metrics such as DSG and VQAScore?
3. Should we expect these patterns to hold for other image generation models?
4. Do the patterns described in the qualitative analysis section hold for other kinds of prompts and concept?

---

> ### Author Response · Authors · 2024-11-14
>
> Thank you for your thoughtful comments. We agree that our discussion could benefit from further strengthening and expansion on specific “details and quirks.” Our primary focus in this work is not so much on the performance of the generated prompts themselves, but rather on evaluating the discrete optimizers. Our goal is to provide a deeper understanding of their respective strengths and weaknesses for image inversion tasks.
>
> As we discuss that CLIP can serve as a limited proxy for alignment between generated images and text in SD-2.1, we believe that prompt quality may vary with different image generation models, depending on the proxy model or loss used. However, we expect the optimizers’ behavior to remain consistent: GCG will continue to find adversarial prompts; PEZ will exhibit strong regularization and transferability; AutoDAN's fluency component will likely retain its advantage by aligning more closely with the learned token input distribution; and Random Search will remain a valuable baseline.
>
> We also agree that more robust metrics, such as DSG or VQAScore, could yield higher-quality prompts. However, we do not anticipate that these would offer new insights into the behavior of the optimizers.
>
> In future revisions, we will adjust our discussion to de-emphasize prompt quality and focus more on the optimizers' behavior, as this is the primary focus of our study.

---

### Meta-Review · Area_Chair_2zez · 2024-12-19

**Metareview:**

The paper presents an empirical analysis of prompt recovery methods in text-to-image models, showcasing both qualitative and quantitative results.

The paper was reviewed by four experts in the domain who acknowledged that the paper was well written and easy to follow (BePT, uWcM, Uzm8, 4nSo), the quantitative analysis considered several evaluation types / metrics (BePT, uWcM), the work was well positioned in the literature (BePT), and the experiments were easy to understand (4nSo).

The main concerns raised by the reviewers were:
1. Missing sufficiently in-depth discussion on the findings and early experimentation of proposed solutions to the observed challenges, making the contribution rather limited (BePT, 4nSo)
2. The use of CLIPscore should be revised to increase the reliability of the observations (BePT)
3. Analysis performed on stable diffusion images only, and the model of choice relying on CLIP, thus limiting its insights and takeaways (BePT, uWcM, Uzm8, 4nSo)
4. Claims not well supported by experimental evidence -- e.g. differences in vocabulary in image captioning and generation models (uWcM)
5. Missing related works in the discussion (uWcM, 4nSo)
6. Unclear significance of the presented results (Uzm8)

During rebuttal and discussion, the authors acknowledge the relevance of some of the points raised by the reviewers and consider those to improve future iterations of their work. Unfortunately, the rebuttal does not take the opportunity to strengthen the analysis and back up claims with additional empirical evidence, as requested by the reviewers. After discussion, the reviewers unanimously recommend rejection. The AC agrees with the reviewers' assessment and recommends to reject. The AC encourages the authors to consider the feedback received to improve future iterations of their work.

**Additional Comments On Reviewer Discussion:**

See above.

---

### Decision · Program_Chairs · 2025-01-22

Reject